# Negative Hyperselection in Metastatic Colorectal Cancer for First-Line Anti-EGFR Therapy: A Narrative Review

**DOI:** 10.3390/ijms26052216

**Published:** 2025-02-28

**Authors:** Giuliana Ciappina, Enrica Toscano, Alessandro Ottaiano, Maurizio Capuozzo, Pierluigi Consolo, Enrica Maiorana, Patrizia Carroccio, Tindara Franchina, Antonio Ieni, Annabella Di Mauro, Massimiliano Berretta

**Affiliations:** 1Division of Medical Oncology, AOU “G. Martino” Hospital, University of Messina, 98125 Messina, Italy; 2School of Specialization in Medical Oncology, Department of Human Pathology “G. Barresi”, University of Messina, 98125 Messina, Italy; 3Istituto Nazionale Tumori di Napoli, IRCCS “G. Pascale”, Via M. Semmola, 80131 Naples, Italy; 4Pharmaceutical Department, Asl Napoli 3 Sud, Marittima Street 3, 80056 Ercolano, Italy; 5Department of Clinical and Experimental Medicine, University of Messina, 98125 Messina, Italy; 6Department of Human Pathology of Adult and Childhood “Gaetano Barresi”, University of Messina, 98121 Messina, Italy

**Keywords:** metastatic colorectal cancer, negative hyperselection, RAS, BRAF, next generation sequencing, liquid biopsy, anti-EGFR treatment

## Abstract

Colorectal cancer (CRC) remains a leading cause of cancer-related mortality, with metastatic disease posing significant therapeutic challenges. While anti-EGFR therapy has improved outcomes for patients with *RAS* and *BRAF* wild-type tumors, resistance remains a major hurdle, limiting treatment efficacy. The concept of negative hyperselection has emerged as a refinement of molecular profiling, identifying additional genomic alterations—such as *HER2* and *MET* amplificationsand *MAP2K1* mutations—that predict resistance to anti-EGFR agents. Studies incorporating these expanded assessments have demonstrated that nearly half of patients with *RAS*/*BRAF* wild-type tumors harbor alternative resistance biomarkers, underscoring the need for expanded selection criteria. Liquid biopsies, particularly circulating tumor DNA (ctDNA) analysis, have revolutionized precision oncology by providing a minimally invasive, real-time assessment of tumor dynamics. ctDNA-based hyperselection enables the detection of resistance-associated alterations, guiding treatment decisions with greater accuracy than conventional tissue biopsies. Recent trials support the predictive value of ctDNA-defined negative hyperselection, revealing superior outcomes for patients stratified through liquid biopsy. This narrative review explores the evolving role of molecular hyperselection in first-line anti-EGFR therapy, emphasizing the integration of ctDNA to refine patient selection, enhance therapeutic efficacy, and pave the way for personalized treatment strategies in metastatic CRC.

## 1. Introduction

Colorectal cancer (CRC) is one of the most common malignancies in the general population. Unfortunately, approximately 20% of cases present with distant metastases at diagnosis, while an additional 30–40% develop metastases following surgical resection of the primary tumor [1,2]. Regarding the treatment of advanced-stage disease, characterized by metastatic dissemination, chemotherapy and novel therapeutic strategies have contributed to a decline in CRC-related mortality over the past decade. In particular, the incorporation of monoclonal antibodies targeting EGFR (epidermal growth factor receptor) and VEGF (vascular endothelial growth factor) into 5-fluorouracil (5-FU)-based chemotherapy has significantly improved both disease-free and overall survival (OS) [3]. Among the clinical and molecular factors considered critical for guiding anti-EGFR therapy, tumor sidedness (as anti-EGFR agents demonstrate greater efficacy in left-sided tumors) and the mutational statuses of *KRAS* (Kirsten rat sarcoma viral oncogene homolog), *NRAS* (neuroblastoma RAS viral oncogene homolog), and *BRAF* (v-raf murine sarcoma viral oncogene homolog B) are paramount. Current guidelines recommend restricting anti-EGFR therapies to patients with wild-type *RAS* and *BRAF* tumors [4]. However, not all patients respond to anti-EGFR treatment despite the absence of mutations in these genes. Nearly one in three patients derives no benefit from these therapies due to either primary or secondary resistance. Thus, despite therapeutic advancements, resistance remains a major clinical challenge, underscoring the need for a more refined molecular characterization of tumors [5]. In recent years, the concept of “negative hyperselection” has emerged, referring to the enrichment of predictive molecular profiling for anti-EGFR response by incorporating additional molecular partners involved in the EGFR pathway (Figure 1). As we will discuss in more detail later, negative hyperselection excludes tumors that, despite being *RAS*/*BRAF* wild-type, harbor molecular alterations associated with resistance to anti-EGFR therapy. These include *HER2* (human epidermal growth factor receptor 2) and *MET* (mesenchymal-epithelial transition factor) amplifications; *NTRK* (neurotrophic receptor tyrosine kinase), *ROS1* (c-ros oncogene 1), *ALK* (anaplastic lymphoma kinase), and *RET* (rearranged during transfection) fusions; *PIK3CA* (phosphoinositide-3-kinase catalytic subunit alpha) exon 20 and *MAP2K1* (mitogen-activated protein kinase kinase 1) mutations; and *PTEN* (phosphatase and tensin homolog) loss. The presence of these alterations activates alternative signaling pathways, circumventing EGFR blockades and rendering the treatment ineffective. This approach aims to optimize therapeutic efficacy by identifying and excluding patients harboring alternative biomarkers [6].

This review examines the available data on negative hyperselection in first-line treatment, as it may play a pivotal role in refining the selection of anti-EGFR-based therapies.

## 2. Methods

A thorough literature review was performed using PubMed/MedLine and Scopus/Elsevier to identify studies on hyperselection in metastatic colorectal cancer (mCRC) treated with anti-EGFR therapy. The search strategy incorporated the terms “colorectal”, “colon”, or “rectal” in conjunction with “metastatic” AND “hyperselection” or “hyper-selection”. Only English-language articles published between 2017 and 2024 were included. Duplicate records were excluded. A total of 56 studies were retrieved. We excluded reviews, case reports, editorials, and studies that did not report response rates or time-to-outcome measures. To ensure comprehensiveness, reference lists of the selected articles were also examined to reduce the likelihood of missing pertinent literature. Only studies investigating first-line treatment strategies were selected and discussed.

## 3. The Role of Negative Hyperselection in Optimizing First-Line Chemotherapy for Metastatic Colorectal Cancer

The challenge in modern oncology lies in refining patient selection to maximize therapeutic benefit. Currently, biomarkers in mCRC primarily function as negative selection tools, excluding *RAS*- and *BRAF*-mutant tumors, rather than positively identifying cases sensitive to EGFR inhibition. In fact, the individual study of genomic alterations, both for negative and positive selection, is complicated by the rarity of these alterations. The PRESSING study (Primary Resistance in *RAS* and *BRAF* Wild-Type Metastatic Colorectal Cancer Patients Treated with Anti-EGFR Monoclonal Antibodies) was a case–control investigation aimed at elucidating the role of negative hyperselection in optimizing patient selection for first-line anti-EGFR therapy in mCRC. Specifically, the study sought to define the negative predictive value of a curated panel of genomic alterations (PRESSING panel) beyond *RAS* and *BRAF* mutations, which are well-established markers of resistance to anti-EGFR monoclonal antibodies. The PRESSING panel encompassed rare but clinically relevant genetic alterations, including *HER2* and *MET* amplifications, *NTRK*/*ROS1*/*ALK*/*RET* rearrangements, and *PIK3CA* exon 20 mutations. By refining molecular selection criteria, this study aimed to enhance the precision of anti-EGFR therapy, ultimately improving treatment outcomes for *RAS* and *BRAF* wild-type mCRC patients [7]. This prospective study evaluated the impact of PRESSING panel alterations on treatment outcomes in *RAS* and *BRAF* wild-type mCRC patients. The study enrolled 94 patients receiving first-line anti-EGFR therapy, categorizing them as either “resistant” (progressing at first assessment) or “sensitive” (achieving a RECIST (Response Evaluation Criteria in Solid Tumors) response or stable disease for at least six months). Targeted next-generation sequencing (T-NGS) identified PRESSING panel mutations in 42.6% of resistant cases compared to 2.1% in sensitive cases (*p* < 0.001), with *HER2* (14.9%) and *MET* (8.5%) amplifications being the most prevalent. Notably, all alterations within the PRESSING panel were mutually exclusive. Expanded sequencing efforts identified additional *RAS* mutations, increasing the total mutation rate to 51.1%. Patients without PRESSING panel alterations demonstrated significantly prolonged progression-free survival (PFS) (median PFS: 6.3 vs. 2.7 months; HR: 0.18, 95% CI: 0.09–0.35, *p* < 0.001), although OS remained comparable between groups (15.2 vs. 17.3 months; HR: 1.05, *p* = 0.876). These findings support the role of PRESSING panel alterations as predictive markers of anti-EGFR resistance rather than purely prognostic indicators. Additionally, tumor location was found to influence response, with right-sided tumors exhibiting a higher frequency of PRESSING alterations (42%) compared to left-sided tumors (20%), correlating with inferior outcomes. This reinforces the well-documented negative prognostic significance of right-sided primary tumors in molecularly hyperselected populations. Interestingly, the study proposed that integrating PRESSING panel analysis into clinical decision-making could exclude nearly 50% of patients likely to exhibit primary resistance, thereby advancing the paradigm of negative molecular hyperselection. The mutual exclusivity of PRESSING panel alterations underscores their potential role as oncogenic drivers. Their association with resistance further highlights their potential as actionable targets for molecularly defined patient subgroups. However, the absence of a control group (not receiving anti-EGFR therapy) prevents definitive conclusions about their predictive versus prognostic value. Nonetheless, the observed differences in treatment sensitivity and PFS—despite no significant OS disparity—reinforce the argument that these alterations primarily dictate response to EGFR blockade.

Another study reported that the integration of the PRESSING panel with primary tumor sidedness provided further refinement in predicting response to first-line anti-EGFR therapy in mCRC [8]. The study found that PRESSING-positive tumors accounted for 24.6% of cases and were associated with a significantly lower overall response rate (ORR) compared to PRESSING-negative tumors (59.2% vs. 75.3%, *p* = 0.030), as well as markedly shorter PFS (7.7 vs. 12.1 months, *p* < 0.001) and reduced two-year OS (48.1% vs. 68.1%, *p* = 0.021). Notably, the negative prognostic impact of PRESSING positivity was comparable to that of right-sided tumor localization, which itself was linked to an inferior ORR (55.2% vs. 74.1%, *p* = 0.037), shorter PFS (8.4 vs. 11.5 months, *p* = 0.026), and a trend toward reduced two-year OS (50.2% vs. 65.1%, *p* = 0.062). These findings underscore the complementary nature of molecular and anatomical factors in determining treatment outcomes. Moreover, while the addition of FU and leucovorin (LV) to panitumumab maintenance conferred a PFS benefit across the study population, this effect was independent of both sidedness and PRESSING status (interaction *p* = 0.293 and 0.127, respectively). However, patients with right-sided or PRESSING-positive tumors who received single-agent panitumumab during maintenance exhibited particularly poor outcomes, with a median PFS of 7.7 months and two-year OS of 38.5% for right-sided tumors and a median PFS of 7.4 months and two-year OS of 47.0% for PRESSING-positive tumors. These results suggest that these subgroups may require alternative therapeutic strategies beyond anti-EGFR monotherapy in maintenance settings. Collectively, the study supports the notion that comprehensive molecular profiling, as facilitated by the PRESSING panel, can refine patient selection for anti-EGFR regimens, potentially guiding more personalized treatment approaches in mCRC.

A translational analysis within the PANDA (a randomized phase II study of first-line FOLFOX (fluorouracil, leucovorin, and oxaliplatin) plus panitumumab versus 5FU plus panitumumab in RAS and BRAF wild-type elderly metastatic colorectal cancer patients) study explored the role of molecular hyperselection in elderly patients receiving panitumumab plus FOLFOX or 5-FU plus leucovorin [9]. In this trial, a slightly modified version of the PRESSING panel, incorporating *MAP2K1* mutations and expanding *PTEN* alterations to include loss of expression was adopted. Patients with tumors bearing at least one “modified PRESSING panel” alteration were included in the “gene-altered” subgroup, whereas those with no alterations were classified as “hyperselected”. Among 183 enrolled patients, 147 were biomarker-evaluable, with 41 (27.9%) classified as “gene-altered” and 106 (72.1%) as “hyperselected”. Molecular hyperselection was independently associated with improved PFS (12.8 vs. 7.6 months; HR 2.08, *p* < 0.001) and OS (29.5 vs. 20 months; HR 1.82, *p* = 0.002). No significant interaction was observed between the treatment arm and hyperselection status. The gene-altered subgroup had lower ORR (51% vs. 71%; *p* = 0.027), confirming that additional genomic alterations negatively impact treatment efficacy. Importantly, primary tumor sidedness did not significantly alter the impact of molecular hyperselection on outcomes, reinforcing the predictive value of this approach independent of anatomical location. The study provides robust validation of molecular hyperselection in a well-defined elderly patient population. The inclusion of *MAP2K1* mutations and broader *PTEN* alterations reflects an adaptive approach to evolving molecular insights.

## 4. Liquid Biopsies and ctDNA in Negative Hyperselection: Implications for Precision Oncology

Traditional tumor tissue profiling remains the gold standard for detecting resistance-associated genetic alterations, but it is hindered by spatial and temporal heterogeneity and the invasive nature of biopsies. Liquid biopsies, particularly circulating tumor DNA (ctDNA) analysis, offer a minimally invasive alternative. In mCRC, ctDNA is detectable in up to 90% of patients, though its utility depends on the sensitivity of sequencing technologies [10]. One of the advantages of liquid biopsies over traditional tissue biopsies, which provide only a static snapshot of a tumor at a single point in time, is their ability to leverage the presence of ctDNA, circulating tumor cells (CTCs), and extracellular vesicles in bodily fluids—primarily blood—to track tumor evolution with unprecedented precision [11]. This dynamic profiling could be particularly valuable in metastatic colorectal cancer (mCRC), where spatial and temporal heterogeneity present significant challenges for treatment selection.

ctDNA, released by tumor cells and isolated from the plasma fraction of the bloodstream, carries the genetic and epigenetic alterations of the tumor, making it a highly informative biomarker. Technological advances in next-generation sequencing (NGS) and digital droplet PCR (ddPCR) have dramatically increased the sensitivity of ctDNA detection, enabling the identification of resistance-associated mutations, minimal residual disease (MRD), and early relapse with high specificity [12,13]. In mCRC, key alterations such as *KRAS*, *NRAS*, and *BRAF* mutations, as well as amplifications in *HER2* and *MET*, can be readily detected in ctDNA, guiding real-time therapeutic decisions. The ability to monitor emerging resistance mutations—such as secondary alterations in *EGFR* that restore downstream signaling—has profound implications for optimizing the sequencing of targeted therapies [14,15].

Beyond genomic profiling, liquid biopsy is now being explored for multi-omic approaches, integrating transcriptomic, proteomic, and epigenetic data to refine predictive and prognostic assessments. The ability to detect hypermethylation patterns or gene fusion events further expands its clinical utility, offering a more comprehensive view of tumor biology [16,17,18]. Despite its promise, liquid biopsy faces challenges, including high costs, the need for standardization, the optimization of sensitivity thresholds, and the interpretation of low-frequency variants. However, as sequencing technologies continue to evolve and artificial intelligence-driven analytics enhance data interpretation, the clinical integration of liquid biopsy is expected to refine precision oncology paradigms, making cancer treatment increasingly personalized and adaptive [19].

Shitara et al. investigated the predictive value of ctDNA in evaluating the efficacy of anti-EGFR therapy through an exploratory biomarker analysis within the phase 3 PARADIGM trial [20]. This analysis utilized the PlasmaSELECT-R 91 PGDx panel, designed to detect 90 mutations, 26 gene amplifications, and threerearrangements in mCRC-related genes, as well as microsatellite instability. Among 733 patients with evaluable baseline ctDNA, those classified as “negative hyperselected” (lacking resistance-associated mutations) experienced superior outcomes with panitumumab plus FOLFOX6 compared to bevacizumab plus FOLFOX6, irrespective of tumor sidedness. In the overall cohort, median OS was 40.7 months versus 34.4 months in favor of the panitumumab-based regimen. “Negative hyperselected” status was defined as the absence of prespecified gene alterations in plasma ctDNA, including mutations in *BRAF* V600E, *KRAS*, *PTEN*, and *EGFR* extracellular domain (ECD) exons 1–16 and *NRAS*, as well as amplifications of *HER2* and *MET* and gene fusions involving *RET*, *NTRK1*, and *ALK*. Conversely, the “gene-altered” category included patients with detectable ctDNA alterations in any of these genes. Notably, ctDNA profiling emerged as a more powerful predictive biomarker than tumor sidedness, challenging conventional treatment stratification paradigms. The superior efficacy of panitumumab plus FOLFOX in “negative hyperselected” patients compared to bevacizumab plus FOLFOX can be attributed to the distinct mechanisms of action of receptor-targeted versus ligand-targeted therapies. Specifically, panitumumab is a monoclonal antibody that directly targets EGFR, whereas bevacizumab targets VEGF, a key ligand involved in angiogenesis. In “negative hyperselected” patients, the absence of resistance-associated mutations (e.g., *KRAS*, *NRAS*, *BRAF* V600E, and *EGFR* extracellular domain mutations) ensures that the EGFR pathway remains a primary driver of tumor progression, making these tumors more susceptible to EGFR blockade. Without alternative activating mutations that could bypass EGFR inhibition, panitumumab effectively suppresses tumor growth by preventing ligand-induced receptor activation. In contrast, bevacizumab neutralizes VEGF, inhibiting angiogenesis and reducing tumor vascular supply. However, VEGF blockade does not directly interfere with tumor-intrinsic signaling pathways, whereas EGFR inhibition does. Consequently, tumors with an intact EGFR pathway may continue to proliferate despite angiogenesis suppression. Thus, while bevacizumab is effective in a broader patient population, its therapeutic benefit is less dependent on specific molecular alterations within the tumor cells.

An exploratory analysis presented at ASCO 2024 by Uetake investigated acquired gene alterations in 276 patients with progressive disease (PD) following anti-EGFR therapy in the phase 3 PARADIGM trial (NCT02394834) [21]. This study integrated solid tumor tissue and ctDNA analysis, utilizing the PlasmaSELECT-R 91 PGDx panel to characterize molecular changes emerging after treatment. In the PARADIGM trial, the addition of panitumumab to mFOLFOX improved OS compared to bevacizumab in patients with *RAS* wild-type mCRC. However, resistance to treatment remains a major challenge, necessitating a deeper understanding of the molecular evolution associated with disease progression. Among the 802 enrolled patients, 390 discontinued therapy due to PD, and 276 (70.8%) had paired pre- and post-treatment ctDNA samples available for analysis. In the panitumumab arm, acquired alterations in the RTK (receptor tyrosine kinase)/RAS pathway (including mutations in *KRAS*, *NRAS*, *PTEN*, and *EGFR* and others such as amplifications and fusions of genes like *HER2*, *MET*, *ALK*, *RET*, and *NTRK1*) were linked to significantly shorter post-progression survival (PPS) (median 13.2 vs. 18.8 months, HR 1.88, 95% CI: 1.28–2.76), with a similar trend observed for OS. In contrast, in the bevacizumab arm, acquired mutations in the CpG island methylator phenotype (CIMP) pathway were associated with worse PPS (median 14.9 vs. 18.6 months; HR 1.58, 95% CI: 1.09–2.29), highlighting distinct mechanisms of resistance between anti-EGFR and anti-angiogenic therapies. Interestingly, co-occurring gene alterations were more frequent in panitumumab-treated patients (52.4% vs. 43.3% in bevacizumab), with a notable increase in *RTK*/*RAS* co-alterations (25% vs. 10%). The presence of multiple *RTK*/*RAS* mutations in the panitumumab arm correlated with significantly reduced PPS, emphasizing the role of tumor evolutionary pressure under targeted therapy. These findings reinforce the importance of molecular hyperselection in refining treatment strategies, optimizing therapeutic sequencing, and informing post-progression management in mCRC.

To assess the utility of ctDNA profiling, baseline plasma samples from 120 patients enrolled in the Valentino study—all with left-sided, RAS/BRAF wild-type, HER2-negative, and microsatellite-stable mCRC treated with first-line panitumumab plus FOLFOX—underwent gene assessment [22]. The analysis of ctDNA was performed using two distinct methodologies: digital PCR (dPCR) via the OncoBEAM RAS CRC Assay (SysmexInostics) and NGS with the Oncomine Colon cfDNA Assay (Ion S5 system). The OncoBEAM Assay is capable of detecting 34 RAS mutations across exons 2, 3, and 4 of *KRAS* and *NRAS* with a sensitivity threshold of 0.01%, while the Oncomine Colon ctDNA panel targets 14 key oncogenic regions, including *APC*, *BRAF*, *EGFR*, *ERBB2*, *GNAS*, *MAP2K1*, *PIK3CA*, *TP53*, with a detection limit of 0.1%. The authors focused their attention on a selection based on ctDNA data, focusing on the status of *RAS* and *PIK3CA*. Interestingly, *RAS* and *PIK3CA* mutations were detected in ctDNA in 10 and 15 cases, respectively, with low concordance to tumor tissue sequencing (31.3% for *RAS* and 47.1% for *PIK3CA*). The presence of *RAS* mutations in baseline ctDNA was significantly associated with ORR according to RECIST (44.4% no response in ctDNA *RAS*-mutated vs. 14.0% no response in ctDNA *RAS* wild-type, *p* = 0.039), whereas *PIK3CA* status was not associated with response. Patients harboring *RAS* mutations in ctDNA had a median PFS of 8 months compared to 12.8 months in wild-type cases (HR: 2.49; 95% CI: 1.28–4.81; *p* = 0.007) and a median OS of 17.1 months versus 36.5 months (HR: 2.26; 95% CI: 1.03–4.96; *p* = 0.042). Similarly, *PIK3CA*-mutated patients experienced inferior PFS (8.5 vs. 12.9 months; HR: 2.86; 95% CI: 1.63–5.04; *p* < 0.001) and reduced OS (21.1 vs. 38.9 months; HR: 2.18; 95% CI: 1.16–4.07; *p* = 0.015). Notably, patients with a high variant allele fraction (VAF ≥ 5%) for *RAS*/*PIK3CA* in ctDNA exhibited the worst prognosis, with a median PFS of 7.7 months compared to 13.1 months in wild-type cases (HR: 4.02; 95% CI: 2.03–7.95; *p* < 0.001) and a median OS of 18.8 months versus 38.9 months (HR: 4.07; 95% CI: 2.04–8.12; *p* < 0.001).

To evaluate the potential of ctDNA for early response assessment in mCRC patients undergoing anti-EGFR therapy, Vidal J. et al. conducted a prospective multicenter study [23]. The study enrolled patients with *RAS* wild-type mCRC receiving first-line chemotherapy plus cetuximab, with sequential liquid biopsies collected at baseline and prior to the third treatment cycle (C3). The analysis of ctDNA was performed using the same methodology as the study by Manca et al. [22], employing dPCR via the OncoBEAM RAS CRC Assay and NGS with the Oncomine Colon cfDNA Assay. Tissue *RAS* assessment was performed according to local laboratory. Trunk mutations—defined as somatic mutations with the highest variant allele frequency (VAF) in plasma—were used as key indicators. Genetic alterations in *RAS*, *BRAF*, *MAP2K1*, and *EGFR*-ECD were classified as resistance-associated mutations. Baseline detection rates for *RAS* mutations in ctDNA differed by method: NGS identified mutations in 8.4% of cases (7/83; median VAF: 6.43%), while dPCR demonstrated higher sensitivity, detecting *RAS* alterations in 11.5% of patients (11/96; median VAF: 0.14%). Plasma–tissue discordances in *RAS* mutations were further investigated, ultimately confirming five true discordant cases (5% of all patients; median VAF: 0.11; range: 0.024–0.38%), aligning with prior studies. The study revealed that combining early changes in trunk mutations with variations in resistance-associated mutations allowed for early prediction of treatment response. Patients classified as “early molecular responders”—characterized by a reduction in trunk mutations and a decline in *RAS*/*BRAF* mutation fractions at C3—achieved significantly longer PFS and higher ORR compared to those exhibiting early molecular progression (median PFS: 18.84 vs. 5.58 months; HR: 0.18, *p* < 0.001; ORR: 77.5% vs. 25%, *p* = 0.008). While no absolute cut-off for ctDNA changes was identified to predict clinical outcomes, any reduction in trunk mutations—rather than a fixed threshold—was associated with enhanced treatment efficacy. These findings support the role of ctDNA dynamics as a promising biomarker for early therapeutic monitoring in mCRC patients receiving anti-EGFR therapy.

A summary of key findings from molecular hyperselection studies is reported in Table 1. These findings suggest that baseline ctDNA profiling provides additional prognostic value beyond standard tumor tissue sequencing, refining the molecular selection of patients most likely to benefit from anti-EGFR therapy. Integrating liquid biopsy into clinical decision-making may enhance patient stratification and optimize therapeutic strategies in mCRC management (Table 2).

## 5. Discussion

The concept of negative hyperselection has emerged as a promising strategy to refine the selection of patients eligible for first-line anti-EGFR therapy in mCRC [6]. The use of comprehensive molecular profiling, particularly through the PRESSING panel, has significantly enhanced our understanding of resistance mechanisms, reinforcing the need for a more tailored therapeutic approach [7]. While current clinical guidelines primarily rely on *RAS* and *BRAF* wild-type status to define anti-EGFR eligibility, the identification of additional genetic alterations—such as *HER2* and *MET* amplifications, *NTRK*/*ROS1*/*ALK*/*RET* fusions, and *PIK3CA* exon 20 mutations—has further stratified patient populations, highlighting subsets with primary resistance to EGFR blockades [7,8,9,20,21,22,23].

One of the major findings supporting negative hyperselection is the significantly lower PFS observed in patients harboring PRESSING panel alterations [7,8,9]. The mutual exclusivity of these alterations underscores their role as independent oncogenic drivers contributing to intrinsic resistance. Furthermore, their association with right-sided tumor location—another well-documented negative prognostic factor—suggests that anatomical and molecular variables must be jointly considered in treatment decisions. The predictive role of the PRESSING panel was validated across multiple studies, demonstrating its potential utility in excluding up to 50% of patients unlikely to benefit from anti-EGFR therapy, thereby sparing them from ineffective treatment and associated toxicities.

Despite these promising findings, several challenges remain and warrant further discussion. The diverse genetic landscape within the “gene-altered” subgroup complicates treatment optimization, as distinct alterations exhibit varying resistance profiles to anti-EGFR therapy. While *HER2* amplification is a well-established driver of resistance, informing the use of HER2-targeted therapies, the clinical relevance of other alterations, such as *RET* fusions, remains insufficiently characterized in mCRC. This highlights the necessity for further research to elucidate the functional consequences of specific mutations and refine EGFR-driven treatment strategies [24]. Another critical aspect of negative hyperselection is the role of liquid biopsy in refining patient stratification. Circulating tumor ctDNA-based analysis has emerged as a powerful, non-invasive tool for real-time molecular profiling, overcoming the spatial and temporal limitations of tissue biopsies. Studies have demonstrated the feasibility of using ctDNA to detect resistance-associated alterations with high sensitivity, enabling dynamic treatment monitoring. However, discordance between liquid and tissue biopsy results, although minimal, necessitates cautious interpretation. The integration of ctDNA-based selection with molecular hyperselection could enhance precision medicine approaches, but large-scale prospective validation is required to establish its clinical applicability. Finally, it is worth emphasizing that the concept of negative hyperselection may extend beyond mCRC to other malignancies in which targeted therapies are employed and resistance mechanisms are well-characterized. This strategy enhances patient selection by excluding tumors that, while lacking common resistance mutations, harbor alternative molecular alterations that drive therapeutic failure. In non-small cell lung cancer, for instance, *EGFR*-mutant tumors with *MET* amplification, *HER2* mutations, or *PIK3CA* alterations frequently develop resistance to EGFR inhibitors [25]. Likewise, in HER2-positive breast cancer, the presence of *PIK3CA* mutations or *PTEN* loss can reduce sensitivity to HER2-targeted agents [26]. A similar phenomenon is observed in *BRAF*-mutant melanoma, where co-occurring *PTEN* loss or *NRAS* mutations activate alternative signaling pathways, ultimately limiting the efficacy of BRAF inhibitors [27,28].

## 6. Conclusions

Negative hyperselection has the potential to transform the therapeutic landscape of mCRC by enhancing patient selection and improving treatment outcomes. Future research should focus on refining molecular stratification strategies, integrating liquid biopsy technologies, and expanding the repertoire of targeted therapeutic options for patients with identified resistance mechanisms. These efforts will be instrumental in advancing personalized medicine and optimizing the efficacy of anti-EGFR therapy in mCRC.

## Figures and Tables

**Figure 1 ijms-26-02216-f001:**
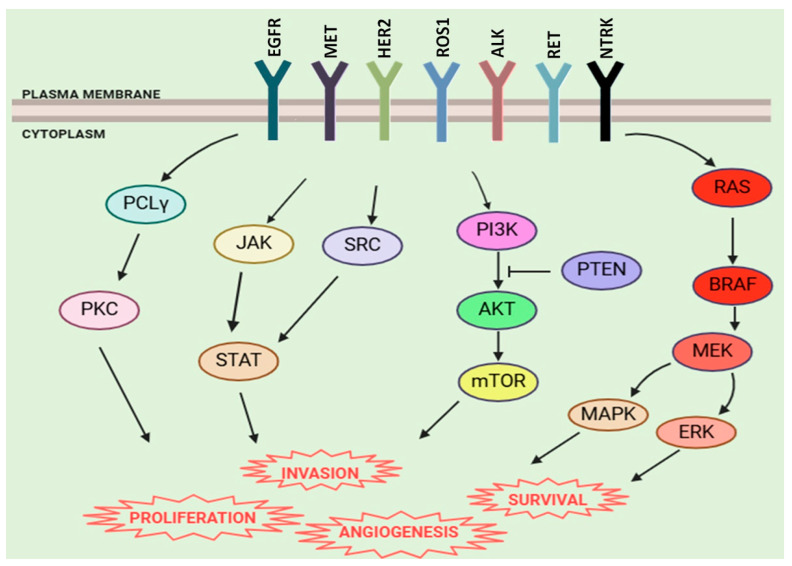
The EGFR pathway exhibits significant redundancy, as many intracellular effectors, including PIK3CA, are shared among multiple receptor tyrosine kinases (RTKs) such as MET, HER2, ROS1, ALK, RET, and NTRK. This functional overlap implies that even in the absence of activating RAS mutations, overexpression or hyperactivation of one of these receptors can sustain pathway activation despite EGFR blockades with monoclonal antibodies (e.g., cetuximab, panitumumab). Consequently, targeting EGFR alone may be insufficient to fully suppress downstream signaling, highlighting the compensatory potential of alternative RTK activation.

**Table 1 ijms-26-02216-t001:** Summary of key findings from molecular hyperselection studies in first-line anti-EGFR treatment of metastatic colorectal cancer.

Study	Population	Methodology	Treatments	Response Rate	Time-to-Outcome
Cremolini et al. [7]	94 RAS/BRAF wt mCRC patients.	PRESSING panel.	FOLFIRI + cetuximab/panitumumab.	Resistance alterations in 42.6% of resistant vs. 2.1% of sensitive patients (*p* < 0.001).	PFS was longer in hyperselected patients compared to those with any gene alteration (mPFS 6.3 vs. 2.7 months). OS was comparable between the groups (15.2 vs. 17.3 months, *p* = 0.876).
Morano F et al. [8]	199 mCRC patients stratified by PRESSING status and tumor sidedness.	PRESSING panel.	FOLFOXIRI + cetuximab/panitumumab.	PRESSING-positive tumors (24.6%) had lower ORR (59.2% vs. 75.3%, *p* = 0.030).	PRESSING-positive tumors had shorter PFS (7.7 vs. 12.1 months, *p* < 0.001) and reduced two-year OS (48.1% vs. 68.1%, *p* = 0.021).
Pietrantonio F et al. [9]	183 elderly mCRC patients (147 biomarker-evaluable).	Modified PRESSING panel (MAP2K1 mutations, broader PTEN alterations).	FOLFOX or 5-FU/LV + panitumumab.	ORR was lower in “gene-altered” patients (51% vs. 71%, *p* = 0.027).	Hyperselected cases (72.1%) had longer PFS (12.8 vs. 7.6 months, *p* < 0.001) and OS (29.5 vs. 20 months, *p* = 0.002).
Shitara et al. [20]	733 mCRCpatients with baseline ctDNA.	ctDNA through PlasmaSELECT-R 91.	FOLFOX + panitumumab or bevacizumab.	In the negative hyperselected population, the ORR was higher with panitumumab (81.5%) than with bevacizumab (66.8%, *p* < 0.001).	“Negative hyperselected” patients had superior OS with panitumumab vs. bevacizumab (40.7 vs. 34.4 months).
Uetake et al. (ASCO 2024) [21]	276 mCRC patients with PD post-anti-EGFR therapy.	ctDNA through PlasmaSELECT-R 91.	Rechallenge with anti-EGFR or switch to bevacizumab-based therapy.	Not available.	Acquired RTK/RAS alterations were associated with shorter PPS (13.2 vs. 18.8 months, HR 1.88, 95% CI: 1.28–2.76) and OS. Similar trend in bevacizumab arm with CIMP alterations.
Manca et al. [22]	120 left-sided, RAS/BRAF wild-type, HER2-negative, and microsatellite-stable mCRC patients.	ctDNA analysis was performed using the OncoBEAM Assay for RAS mutations and the Oncomine Colon panel. The results were compared with tumor tissue sequencing using the PRESSING panel.	FOLFOX + panitumumab.	44.4% no response in ctDNA RAS-mutated vs. 14.0% no response in ctDNA RAS wild-type, *p* = 0.039. No significance for PIK3CA.	A high variant allele fraction (VAF ≥ 5%) for RAS/PIK3CA mutations in ctDNA exhibited a median PFS of 7.7 months vs. 13.1 months in wild-type cases (HR: 4.02; 95% CI: 2.03–7.95; *p* < 0.001) and a median OS of 18.8 months vs. 38.9 months (HR: 4.07; 95% CI: 2.04–8.12; *p* < 0.001).
Vidal et al. [23]	99 RAS wt mCRC patients receiving first-line treatment.	ctDNA analysis was performed using the OncoBEAM Assay for RAS mutations and the Oncomine Colon panelat baseline and prior to the third treatment cycle. RAS tumor tissue was assessed according to local laboratory.	Chemotherapy + cetuximab.	Early molecular responders showed significantly better ORR compared to early molecular progression (ORR: 77.5% vs. 25%, *p* = 0.008).	Early molecular responders showed significantly longer PFS compared to early molecular progression (median PFS: 18.84 vs. 5.58 months; HR: 0.18, *p* < 0.001).

CIMP: CpG island methylator phenotype; CI: confidence interval; ctDNA: circulating tumor DNA; EGFR: epidermal growth factor receptor; FOLFIRI: 5-fluorouracil, leucovorin, and irinotecan; FOLFOX: 5-fluorouracil, leucovorin, and oxaliplatin; FOLFOXIRI: 5-fluorouracil, leucovorin, oxaliplatin, and irinotecan; HR: hazard ratio; mCRC: metastatic colorectal cancer; ORR: objective response rate; OS: overall survival; PD: progressive disease; PFS: progression-free survival; PRESSING: Prospective Molecular Screening for Response to EGFR Inhibition in Metastatic Colorectal Cancer.

**Table 2 ijms-26-02216-t002:** Comparison of tumor tissue and liquid biopsy for molecular hyperselection.

Feature	Tumor Tissue Profiling	Liquid Biopsy (ctDNA)
Invasiveness	High (requires biopsy)	Low (blood-based)
Spatial/temporal heterogeneity	Yes (single site sample)	No (captures systemic mutations)
Detection sensitivity	High (direct tumor sequencing)	Variable (depends on ctDNA shedding)
Application in clinical trials	Well-established	Emerging, requires further validation

## Data Availability

No new data were created.

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
