# Peer review of "Negative Hyperselection in Metastatic Colorectal Cancer for First-Line Anti-EGFR Therapy: A Narrative Review"

_ijms, 2025, doi:10.3390/ijms26052216_

Round 1

Reviewer 1 Report

Comments and Suggestions for Authors

Dear Authors,

I am pleased to review your manuscript, but there are a few issues that need to be addressed:

  1. I came across a paper published in 2024 titled "The role of negative hyperselection in metastatic colorectal cancer." I noticed this reference is not included in your manuscript. Could you confirm whether you cited it? Additionally, could you clarify how your study differs from theirs?

  2. The definition of negative hyperselection seems to be missing from the introduction. While there is a correlation with Figure 1 (lines 64-66), I believe it would be helpful to provide a more detailed explanation for a clearer review. I did find the definition in lines 213-217.

  3. Is Figure 1 adapted from a journal, or was it created using a specific tool? Please provide appropriate acknowledgment or citation.

  4. While your focus is on mCRC, does the concept of negative hyperselection apply to other types of tumors as well? It would be valuable to discuss this in your manuscript.

  5. Table 1 appears somewhat disorganized. It would benefit from some reworking for clarity and structure.

  6. In the discussion section, no references cited throughout. Please check. 

Author Response

Comment 1: I came across a paper published in 2024 titled "The role of negative hyperselection in metastatic colorectal cancer." I noticed this reference is not included in your manuscript. Could you confirm whether you cited it? Additionally, could you clarify how your study differs from theirs?

Response: Please note that the work titled "The Role of Negative Hyperselection in Metastatic Colorectal Cancer" is an editorial on the article by Stahler et al., which presents an analysis of patients receiving maintenance therapy. The editorial is appropriately brief and does not provide molecular insights, detailed clinical data, or comments on the role of liquid biopsy. Moreover, Stahler et al. explore the concept of negative hyperselection to guide the continuation of anti-EGFR therapy in the maintenance setting. Since the original study (not the editorial) does not formally select patients for first-line treatment, it was not included in our narrative review. We hope this clarification provides sufficient details for the Reviewer.

Comment 2: The definition of negative hyperselection seems to be missing from the introduction. While there is a correlation with Figure 1 (lines 64-66), I believe it would be helpful to provide a more detailed explanation for a clearer review. I did find the definition in lines 213-217.

Response: We apologize for this. To enhance clarity for the reader, we have added a definition of "negative hyperselection" in the introduction, providing a clearer understanding of the concept and encouraging further engagement with the text.

Comment 3: Is Figure 1 adapted from a journal, or was it created using a specific tool? Please provide appropriate acknowledgment or citation.

Response: Figure 1 is original and was created using PowerPoint, primarily by three co-authors (Giuliana Ciappina, Alessandro Ottaiano, and Maurizio Capuozzo). Therefore, no acknowledgment or citation is required.

Comment 4: While your focus is on mCRC, does the concept of negative hyperselection apply to other types of tumors as well? It would be valuable to discuss this in your manuscript.

Response: We sincerely thank this reviewer for their scientifically elegant comment, which enhances the informativeness and interest of our review without making it overly complex. In response, we have added a few paradigmatic examples of this concept, along with relevant references, at the end of the Discussion. Please see the revised version, where the changes are highlighted in yellow.

Comment 5: Table 1 appears somewhat disorganized. It would benefit from some reworking for clarity and structure.

Response: The table has been thoroughly revised, completed, and reorganized, and we believe it is now clearer. We have expanded and separated some information while removing details that can be found in the main text. We sincerely thank the reviewer for their valuable feedback, which encouraged us to improve the table.

Comment 6: In the discussion section, no references cited throughout. Please check. 

Response: Some citations are repetitions of previously mentioned studies; however, we agree that including them is appropriate. They have been added. Thank you

Reviewer 2 Report

Comments and Suggestions for Authors

The narrative review titled “Negative Hyperselection in Metastatic Colorectal Cancer for First-Line Anti-EGFR Therapy” by Giuliana Ciappina et al. highlights the significance of Negative Hyperselection over genetic alterations in metastatic colorectal cancer. This review address why this approach is crucial for addressing chemo resistance and utilizing PRESSING panel and tumor-sidedness as a predictive biomarker for selecting anti-EGFR therapy. Additionally, the review explores the advantages of using liquid biopsies for molecular profiling to tackle the limitations associated with tissue-based testing.

Comments are

  • The review provides a lot of information. However, please simplify it so that readers can follow it easily.
  • Provide abbreviations for the terms used.
  • Briefly describe what is uniquely discussed in this review compared to other reports, such as:

Ji, Jingran, and Marwan Fakih. "The role of negative hyperselection in metastatic colorectal cancer." Journal of Gastrointestinal Oncology 15.5 (2024): 2353.

Shitara, Kohei, et al. "Baseline ctDNA gene alterations as a biomarker of survival after panitumumab and chemotherapy in metastatic colorectal cancer." Nature Medicine 30.3 (2024): 730-739.

  • Describe a narrative of the findings in the NCBI/MEDLINE search, including the search terms used. Explain how the results were filtered.
  • The authors may cite the following article, which explains the terminologies for the EGFR pathway:

Sundar, Raghav, Iain Bee Huat Tan, and Cheng E. Chee. "Negative predictive biomarkers in colorectal cancer: PRESSING ahead." Journal of Clinical Oncology 37.33 (2019): 3066-3068.

  • Describe the PRESSING panel in line 99.
  • In lines 179 to 186, describe at what stage of the tumor circulating tumor cells are detected. Briefly summarize the results of dynamic profiling obtained in earlier studies.
  • In line 187, mention which portion of the bloodstream ctDNA is detected in.(serum or plasma)
  • In lines 192 to 193, does ctDNA itself indicate whether there are mutations in KRAS, NRAS, and BRAF, as well as amplifications in HER2 and MET, or is a comparison with tissue biopsy required to confirm the mutations?
  • Explain why panitumumab plus mFOLFOX6 treatment favored negative hyperselection compared to bevacizumab. Discuss this in terms of ligand versus receptor-targeted treatment.
  • For the heading of Table 1, re-frame the heading so that it describes the table is based on ctDNA analysis.
  • If relevant, mention the similarities of hyperselection factors between right- and left-sided mCRC.
  • Some lines are redundant. The authors can avoid repetition.
  • For lines 333 and 334, briefly state what has been discussed in the review.
  • In the discussion section, cite references that summarize resistance-associated mutations described in the text.

Comments on the Quality of English Language

Use simple language for readers to follow.

Author Response

Comment 1: The review provides a lot of information. However, please simplify it so that readers can follow it easily.

Response: One of the key features of this review is its in-depth analysis of the outcomes reported in the few available studies, which have already been summarized. Further condensing this scientific information would risk diminishing the value and purpose of this review, which is to engage both molecular biologists and clinicians. With minimal iconography, we aim to provide a clear, accurate, and compelling understanding of the concept of "negative hyperselection," its significance, and its clinical impact. However, where possible, redundancies have been removed, and the text has been reorganized—particularly in the final section on the use of liquid biopsy.

Comment 2: Provide abbreviations for the terms used.

Response: Throughout the manuscript, we have carefully ensured the inclusion of relevant abbreviations.

Comment 3: Briefly describe what is uniquely discussed in this review compared to other reports, such as: Ji, Jingran, and Marwan Fakih. "The role of negative hyperselection in metastatic colorectal cancer." Journal of Gastrointestinal Oncology 15.5 (2024): 2353. Shitara, Kohei, et al. "Baseline ctDNA gene alterations as a biomarker of survival after panitumumab and chemotherapy in metastatic colorectal cancer." Nature Medicine 30.3 (2024): 730-739.

Response: Please note that the first paper is an editorial, which is appropriately brief. It does not provide molecular insights or detailed clinical data, nor does it offer commentary on the role of liquid biopsy. Furthermore, Stahler et al. explore the role of negative hyperselection in guiding the continuation of anti-EGFR therapy in a "maintenance setting." Since this paper (not the editorial but the study discussed) does not formally select a first-line treatment, it was not included in our narrative review. We hope this explanation is sufficient for the reviewer. The second paper by Shiatara et al. is one of the articles we analyze and discuss in detail.

Comment 4:

Describe a narrative of the findings in the NCBI/MEDLINE search, including the search terms used. Explain how the results were filtered.

Response: In response to the reviewer's request, we recognize the importance of transparency in the literature selection process. However, since this is a narrative review rather than a systematic review, providing a formal flowchart for study selection would be excessive and contextually inappropriate. Narrative reviews are intended to offer a critical synthesis of the literature, emphasizing expert interpretation over a structured, meta-analytical approach. Nonetheless, we have enhanced the methodological description to provide greater clarity. The objective of this narrative review is to synthesize existing evidence, contextualize the findings within the broader field, and highlight key insights without the methodological constraints of a systematic review.

Comment 5: The authors may cite the following article, which explains the terminologies for the EGFR pathway: Sundar, Raghav, Iain Bee Huat Tan, and Cheng E. Chee. "Negative predictive biomarkers in colorectal cancer: PRESSING ahead." Journal of Clinical Oncology 37.33 (2019): 3066-3068.

Response: Thank you for your suggestion. The manuscript has been cited accordingly.

Comment 6: Describe the PRESSING panel in line 99.

Response: It has been done. Please see the revised version.

Comment 7: In lines 179 to 186, describe at what stage of the tumor circulating tumor cells are detected. Briefly summarize the results of dynamic profiling obtained in earlier studies.

Response: The reviewer’s observation is accurate and well-founded. Unfortunately, these studies do not provide "dynamic" assessments but only baseline evaluations, meaning just before the patients start therapy. Only one study includes an assessment at the third chemotherapy cycle. Therefore, this issue cannot be sufficiently analyzed with the available data. It is clear that this is a limitation of the studies, and it is also evident that in the future, repeated and periodic (dynamic) evaluations will certainly be desirable, given the biological and genetic nature of cancer, as they would provide more informative insights.

Comment 8: In line 187, mention which portion of the bloodstream ctDNA is detected in.(serum or plasma)

Response: The clarification has been carried out accordingly. Circulating tumor DNA (ctDNA) is typically detected in the plasma component of the bloodstream. Plasma is preferred over serum for ctDNA extraction because it carries a reduced risk of contamination from genomic DNA originating from leukocytes, which may occur in serum following clotting.

Comment 9: In lines 192 to 193, does ctDNA itself indicate whether there are mutations in KRAS, NRAS, and BRAF, as well as amplifications in HER2 and MET, or is a comparison with tissue biopsy required to confirm the mutations?

Response: This is an ongoing debate, and it is challenging to explore this topic in depth within the scope of the current study. It is, however, a well-established issue among researchers. We would like to engage with the Reviewer on this matter by highlighting that liquid biopsy, which analyzes ctDNA, has been proposed as a non-invasive method for detecting genetic alterations in metastatic colorectal cancer (and other malignancies). While the concordance between ctDNA and tissue biopsy in detecting these alterations is generally high, discrepancies may arise due to factors such as tumor heterogeneity and the sensitivity of ctDNA detection techniques. Thus, while ctDNA profiling provides valuable insights, tissue biopsy remains the gold standard for definitive mutation confirmation. However, we would like to emphasize that delving deeper into this topic and exploring these methodological details is beyond the scope of our work.

Comment 10: Explain why panitumumab plus mFOLFOX6 treatment favored negative hyperselection compared to bevacizumab. Discuss this in terms of ligand versus receptor-targeted treatment.

Response: We thank the Reviewer for encouraging us to expand on these aspects. The study by Shitara et al. has been discussed in line with the Reviewer’s suggestions. Please refer to the revised version, where the additions are highlighted in yellow.

Comment 11: For the heading of Table 1, re-frame the heading so that it describes the table is based on ctDNA analysis.

Response: Table 1 has been improved and fully restructured. Please note that you can identify which studies analyze ctDNA from the "Methodology" column.

Comment 12: If relevant, mention the similarities of hyperselection factors between right- and left-sided mCRC.

Response: Thank you for the suggestion. The text already explains that, in general, "PRESSING" positive tumors are more frequently found in right-sided cancers.

Comment 13: Some lines are redundant. The authors can avoid repetition.

Response: The text has been revised, and some minor lexical adjustments have made it less redundant.

Comment 14: For lines 333 and 334, briefly state what has been discussed in the review.

Response: Thank you for your comment. This section has been revised for clarity. Please refer to the updated version.

Comment 15: In the discussion section, cite references that summarize resistance-associated mutations described in the text.

Response: It has been done in accordance with the Reviewer’s observation, thank you.